# Plough Tillage Maintains High Rice Yield and Lowers Greenhouse Gas Emissions under Straw Incorporation in Three Rice-Based Cropping Systems

Frederick Danso [1], Oluwaseyi Oyewale Bankole [1], Nan Zhang [1], Wenjun Dong [2], Kun Zhang [3], Changying Lu [4], Ziyin Shang [1], Gexing Li [5], Aixing Deng [1], Zhenwei Song [1], Chengyan Zheng [1], Jun Zhang [1,*] and Weijian Zhang [1]

1 Institute of Crop Sciences, Chinese Academy of Agricultural Sciences/Key Laboratory of Crop Physiology & Ecology, Beijing 100081, China; dansotodanso@gmail.com (F.D.); bankole623@gmail.com (O.O.B.); zhangnan@caas.cn (N.Z.); shangziyin@caas.cn (Z.S.); songzhenwei@caas.cn (Z.S.); zhengchengyan@caas.cn (C.Z.); zhangweijian@caas.cn (W.Z.)
2 Cultivation and Farming Research Institute, Heilongjiang Academy of Agricultural Sciences, Harbin 150086, China; dongwenjun0911@163.com
3 National Engineering and Technology Research Center for Red Soil Improvement, Jiangxi Institute of Red Soil, Nanchang 331717, China; zhkp1984@163.com
4 Institute of Agricultural Sciences in Taihu Lake District, Jiangsu Academy of Agricultural Sciences, Suzhou 215100, China; luchangying@163.com
5 College of Agriculture, Henan Agricultural University, Zhengzhou 450046, China; 15993023318@163.com
* Correspondence: zhangjun@caas.cn

**Abstract:** Straw incorporation promotes rice productivity and soil fertility. However, the effects of tillage practice with straw on GHG emissions in paddy fields are not well documented. Under three rice-based cropping systems of China (single rice, double rice and rice-wheat cropping systems), we investigated rice yield, $CH_4$, $N_2O$, area and yield-scaled emissions arising from different straw-incorporated tillage patterns. Tillage with straw affected rice yield by $-6.8 \sim 3.2\%$, $-9.1 \sim 9.0\%$ and $-9.8 \sim 2.1\%$ in single rice, rice-wheat and double rice cropping systems respectively. Straw impacted $CH_4$ emission but tillage influenced its impact irrespective of the rice cropping system. The highest $CH_4$ emissions occurred in RedT + S, RoT + S and RoT + S under single rice, rice-wheat and double rice cropping systems respectively. Cumulative $CH_4$ emission of PT + S decreased by 46.8% ($p < 0.05$) compared to RedT + S in the single cropping system, while under rice-wheat and double rice cropping systems, cumulative $CH_4$ emission of PT + S decreased by 19.0% ($p < 0.05$) and 13.2% ($p > 0.05$) respectively compared with RoT + S. Lower methanogenic abundance of PT + S translated into the lowest cumulative $CH_4$, area and yield scaled emissions in single rice and double rice cropping systems. To maintain high rice yield and reduce GHG emissions from straw incorporation, PT + S is recommended for a rice-based cropping system.

**Keywords:** rice; soil tillage; crop straw; greenhouse gas; methane





## 1. Introduction

Crop production generates a substantial amount of straw, an important component of agricultural residue. With a projected world population of nine billion by 2050, a 50% increase in food production over that presently produced may be needed to meet future demands [1]. This projected increase will exponentially raise the amounts of crop residue generated annually. Due to the historical antecedence of air pollution arising from crop residue burning, sustainable ways to manage the residue are being sort. With only 9.81% of crop residue being returned to croplands and more than 20% of crop residue burnt directly in the field or thrown away [2], it is therefore important to effectively manage its utilization in agricultural systems. To exploit the substantial quantities generated yearly,

crop residue is returned to the field using different tillage methods. As an important agricultural management, tillage influences the soil's physical properties and nutrient distribution in rice fields and also plays a key part in GHG emissions by accelerating crop residue decomposition before rice transplanting [3,4].

Tillage effects on $CH_4$ and $N_2O$ emissions are not always consistent among different studies [5]. Conventional and rotary tillage increased $N_2O$ emissions by 2.3% to 30.8% and −2.3% to 47.4%, respectively [6]. Compared to plough tillage, no tillage decreased cumulative $CH_4$ emissions by 10.1% while rotary tillage increased $CH_4$ emissions by 11.4% [7]. Reduced tillage compared to conventional tillage minimized the global warming potential (GWP) of $CH_4$ and $N_2O$ by 20.8% under the single crop system though not significant under the double crops rotation system [8]. Therefore detailed knowledge of the different tillage methods and straw incorporation in each cropping system on GHG emissions is imperative for the recommendation of low GHG emission practices.

Mitigating $CH_4$ emissions is important, as rice paddies are major $CH_4$ emission sources, contributing about 10–20% of anthropogenic $CH_4$ emissions annually [9]. Thus, $CH_4$ emissions reduction while sustaining high rice yield is important in mitigating global warming. The adoption of tillage can bury crop straw in deeper soil layers, reduce decomposition and thus mitigate $CH_4$ emission [10]. Nonetheless, crop residue use as a nutrient source can be problematic due to its slow decomposition rate [11]. Owing to the minimal synchronization between plant nutrient requirements and nutrient release from straw, incorporating straw into the field can also intensify crop yield reduction [12]. Its residual presence in the soil can hamper root penetration [13], inhibit rice shoot growth after transplanting and impact the paddy's production [11].

However, tillage frequently combined with straw, is considered one of the important ways for soil quality improvement and reduction in environmental consequences arising from the burning of straw [14]. Under straw-incorporated rotary tillage, nitrogen content was more in the 0–10 cm and less in the 20 cm soil depth, as compared to straw-incorporated plough tillage [15]. Straw incorporation using plough tillage produced a lower dissolved organic carbon and total organic carbon in the 0–7 cm and 7–14 cm soil depth, respectively. However, in the 14–21 cm soil depth, higher dissolved organic carbon was noted in straw-incorporated plough tillage than in rotary tillage [16]. Straw retention, besides directly impacting $CH_4$ emissions, also has an overall effect on greenhouse from rice farming system. [17].

As tillage with straw incorporation is widely encouraged across China, it is likely this adoption increases $CH_4$ emissions in China's paddy fields [18]. Therefore an appreciation of the integrated effects of different soil tillage, and straw on GHG emission and rice yield is important in the selection of crop straw retention patterns for high yield and less GHG emission. Owing to different tillage methods, and varying climatic and rice cropping systems across China, the single rice, rice-wheat and double rice cropping systems have important implications for tillage with straw incorporation in paddies. The objectives of this study are (a) to evaluate the integrated impacts of different tillage methods with straw incorporation on rice yield and GHG emission, and (b) to recommend good methods of crop straw incorporation for high-yielding and less GHG emission straw in China's rice cropping systems.

## 2. Materials and Methods

### 2.1. Experimental Location

The study was conducted in 2019 at three major rice cropping systems in China namely: single rice cropping at the National Modern Agricultural Demonstration Park, Minzhu town, Heilongjiang Province (45°49 N, 126°48′ E,) on a Chernozem soil (a Mollisols in USA-ST); rice-wheat cropping at the Jiangsu Modern Agricultural Demonstration Park in Taican town, Jiangsu Province (31°33′50 N, 121°10′26 E) on a Fluvisols soil and double rice cropping at the Jiangxi Institute of Red Soil, Jinxian, Jiangxi Province (28°37′ N, 116°26′ E) on a Stagnic Anthrosols soil. The single rice cropping system is dominated by a northern temperate climate with mean annual precipitation of 508–583 mm, effective accumulated temperature (≥10 °C) 26–27 °C, annual sunshine hours of 2668 and 131–146 days frostless

period. The rice-wheat cropping system is dominated by a hot and temperate climate, with annual mean precipitation of 1070 mm and an annual mean temperature of 15.6 °C. The double rice cropping is characterized by a subtropical climate, an annual average precipitation of 1537 mm with a mean of 262 frost-free days a year and an annual average temperature of 18.1 °C. The cropping cycle of the double rice cropping focused on only the late rice season. The weather data in the three cropping systems and the initial soil properties determined are shown in Figure 1 and Table 1 respectively.

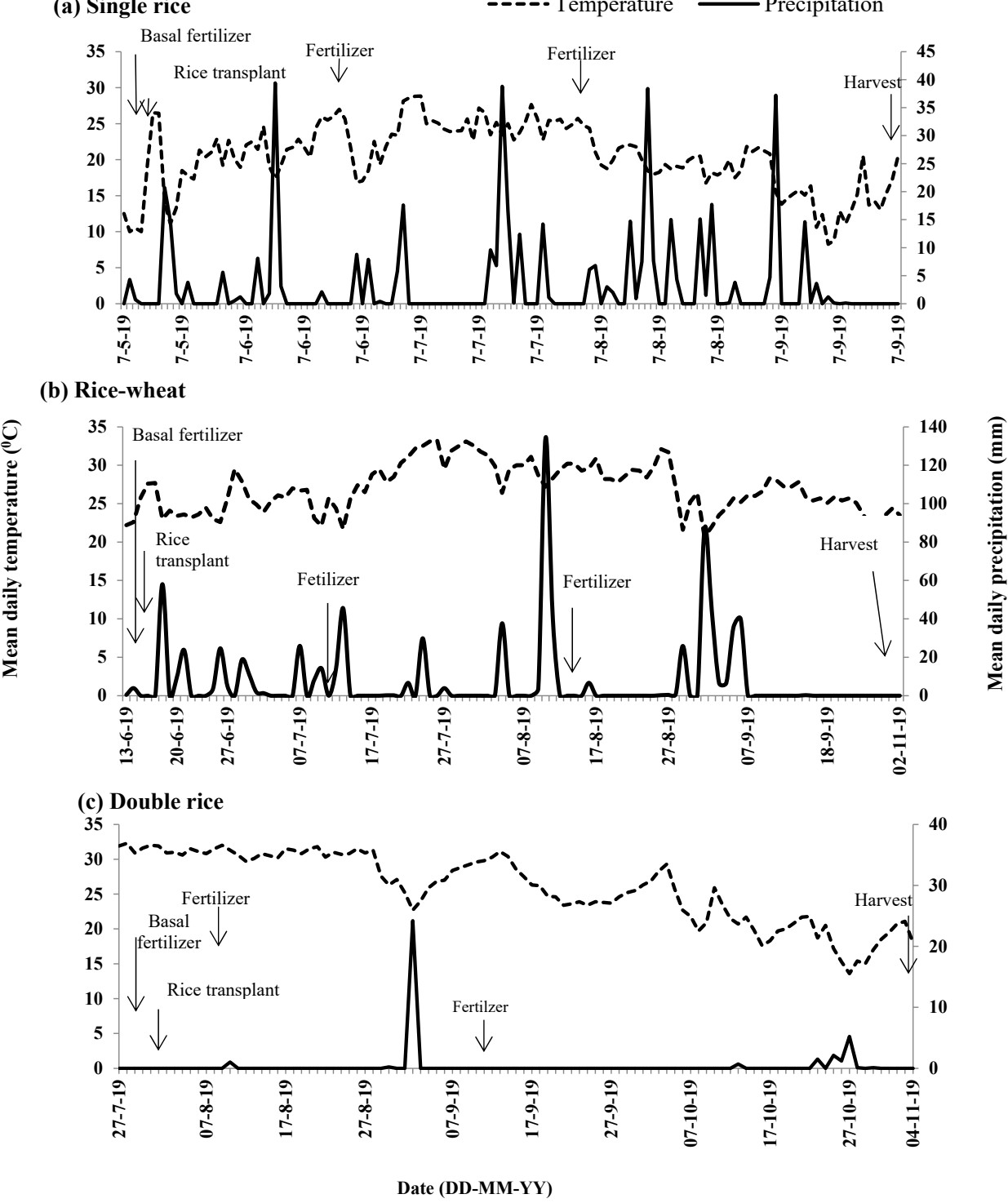

**Figure 1.** Daily average temperature, precipitation and management practices during the rice growing seasons in the three cropping systems.

**Table 1.** Weather conditions and initial soil properties.

| Cropping System | Average Air Temperature (°C) | Total Precipitation (mm) | pH | Soil Organic Matter (g kg$^{-1}$) | Available N (mg kg$^{-1}$) | Available P (mg kg$^{-1}$) | Available K (mg kg$^{-1}$) |
|---|---|---|---|---|---|---|---|
| Single rice | 19.4 | 739.2 | 8.6 | 27.3 | 78.9 | 24.2 | 184.7 |
| Rice-wheat | 27.2 | 781.5 | 6.8 | 26.1 | 135.3 | 28.6 | 92.2 |
| Double rice | 25.9 | 350.2 | 5.9 | 22.6 | 128.5 | 57.2 | 100.3 |

*2.2. Experimental Design*

Adopting a randomized complete block design with three replication and plot sizes of 15 m × 10 m, the tillage methods in the various locations were imposed as: in single rice cropping, plough tillage without straw incorporation (PT − S), reduced tillage with straw incorporation (RedT + S), rotary tillage with straw incorporation (RoT + S), plough tillage with straw incorporation (PT + S) and in both rice-wheat and double rice cropping systems, rotary tillage without straw incorporation (RoT − S), reduced tillage with straw incorporation (RedT + S), rotary tillage with straw incorporation (RoT + S), plough tillage with straw incorporation (PT + S). Since the baseline of soil tillage was different, PT − S and RoT − S were respectively implemented as the control in different test sites. Rice straw was used in both the single and double rice cropping systems while wheat straw was adopted in the rice-wheat cropping system. In reduced tillage treatments, straw was incorporated manually on the soil surface at a depth of 3.0–5.0 cm, allowed to imbibe water and sink below the water surface. Straw was incorporated in rotary tillage at a depth of 15 cm and plough tillage at depth of 30 cm using tractor-mounted ploughs in all three cropping systems.

Straw was incorporated into the different tillage treatments at rates of 7.0 t ha$^{-1}$ with 180 kg N ha$^{-1}$, 75 kg $P_2O_5$ ha$^{-1}$, 60 kg $K_2O$ ha$^{-1}$ in the single cropping system; 7.5 t ha$^{-1}$ with 270 kg N ha$^{-1}$, 75 kg $P_2O_5$ ha$^{-1}$, 60 kg $K_2O$ ha$^{-1}$ in rice-wheat cropping system; 7.0 t ha$^{-1}$ with 195 kg N ha$^{-1}$, 75 kg $P_2O_5$ ha$^{-1}$, 60 kg $K_2O$ ha$^{-1}$ in the double rice cropping system. All fertilizers were manually broadcast into each plot. Nitrogen fertilizer in the form of urea was used as 50% N basal fertilizer before transplanting, 30% N at rice tillering, and 20% N during panicle initiation in the single rice and rice-wheat cropping. In the double rice, 50% of the total N was applied as a basal fertilizer before transplanting, 20% N at mid tillering, and 30% N at the panicle initiation stage during the rice season.

Based on the local management practices for high rice yield in both single rice and double rice cropping systems, fields were submerged with water for 3–5 days, puddled before sowing and maintained under continuous flooding to meet the soil moisture conditions required for transplanting. In the rice-wheat cropping system, land preparation was carried out by tilling and leveling the soil under dry conditions. Adopting the dry direct-seeding method, dry rice seeds were sown and maintained in moist conditions to allow for rice emergence. Subsequently, fields were irrigated using alternate wet and dry irrigation methods in line with the local management strategy. Nursed local japonica rice seeds (Longjing 21 in single rice) and local indica rice seeds (Taiyou 871 in double rice) were transplanted manually at three seedlings hill$^{-1}$ and 25 hills m$^{-2}$. In the rice-wheat cropping, RedT + S was planted with Nanjing 44 rice seeds using the dry direct seeding method at a seeding rate of 4 kg ha$^{-1}$ while the other treatments were machine transplanted at three seedlings hill$^{-1}$ and 25 hills m$^{-2}$.

*2.3. Greenhouse Gas Sampling and Determination*

The static closed chamber and gas chromatography methods were adopted to sample and measure $CH_4$ and $N_2O$ [19]. Polyvinyl chloride (PVC) chambers with dimensions (length × width × height) 50 cm × 50 cm × 50 cm or 50 cm × 50 cm × 100 cm per the rice height and equipped with battery-operated air circulating fans to ensure complete gas mixing in the headspace was used. To avoid the fertilizer-chasing effects, greenhouse gas sampling dates were chosen not to coincide with the date of fertilizer application. From 9:00 to 11:00 am weekly, the chamber headspace gas was sampled into pre-evacuated

vacuum tubes via a 50 mL airtight syringe equipped with a three-way stop-cork at intervals of 0, 5, 10 and 15 min after chamber closure while an attached digital thermometer was used to read chamber temperature. Collected gases were analyzed to obtain $CH_4$ and $N_2O$ concentrations using a gas chromatograph (Agilent 7890A, Agilent Technologies, Wilmington, DE, USA) equipped with a flame ionization detector (FID) and an electron capture detector (ECD) to detect $CH_4$ and $N_2O$ respectively. The slope of the mixing ratio of four sequential samples was used in the determination of both $CH_4$ and $N_2O$ fluxes [9,19]. Cumulative $CH_4$ and $N_2O$ emissions were calculated using the formula described by [20].

$$Cumulative\ (CH_4\ or\ N_2O)\ emission = \sum_{i=1}^{n}(B_i + B_{i+1})/2 \times (t_{i+1} - t_1) \times 24$$

where $B$ is the $CH_4$ or $N_2O$ flux (mg m$^{-2}$ h$^{-1}$), $i$ is the *ith* measurement, the term $(t_{i+1} - t_i)$ is the difference of two adjacent measurements of time in days, and $n$ is the number of measurements in total. Based on a 100-year time horizon and a radiative forcing potential of 27.9 for $CH_4$ and 273.0 for $N_2O$ [21], the area-scaled GHG emission was calculated in $CO_2$ equivalent ($CO_2$-eq) as follows:

$$Area\text{-}scaled\ GHG\ emission\ (kg\ CO_2\text{-}eq\ ha^{-1}\ yr^{-1}) = 27.9 \times CH_4 + 273 \times N_2O$$

where $CH_4$ and $N_2O$ represent the seasonal cumulative emissions.

Yield-scaled GHG emission was computed by dividing area-scaled emission by rice yield [22].

### 2.4. Measurement of Rice Yield

Three replicates of one m$^2$ rice plant at maturity of each treatment were harvested using one m$^2$ quadrant. Rice plant that fell within the one m$^2$ quadrant was harvested for yield determination. Grain yields were adjusted at 14.5% and 13.5% moisture content for japonica and indica rice respectively.

### 2.5. Soil Sampling and Determination

Three replicates of initial soil samples collected at a depth of 0–15 cm per plot, were composited, air-dried and ground to pass through a 2 mm sieve. Total Nitrogen (TN) was determined from ground air-dried samples by the dry combustion method using an Elemental Analyzer (Elementar Analysen system GmbH, Hanau, Germany). Soil-available phosphorus (AP) was extracted with 0.5 mol L$^{-1}$ NaHCO$_3$ and evaluated calorimetrically [23], whereas soil available potassium (AK) was analyzed by flame photometry after extraction with 1M ammonium acetate [24]. Soil pH was determined using the pH meter. At 0–15 cm soil depth in each treatment, soil samples were collected at three randomized auger points at the rice tillering stage (3rd week in rice-wheat, 1st week in double rice cropping and 5th week in single rice). The ten samples collected from every field were pooled and mixed thoroughly into a composite sample.

Collected composite soil samples were placed in an ice pack container for onward transportation and storage in a freezer. The portions of the composite soil samples were stored at $-80\ °C$ for molecular microbial assay. Methyl coenzyme M reductase (*mcrA*) and particulate methane monooxygenase (*pmoA*) genes copy numbers representing methanogenic and methanotrophic abundance were quantified using the Line-Gene 9600 Plus Real-time PCR system (Bioer, Hangzhou, China), with the primer pair MLf (5′-GGTGGTGTMGGATTC ACACAR-TAYGC WACAGC-3′) and MLr (5′-TTCATTGCRTAGTTWGGRTAG TT-3′) [25], and A189F (5′-GGNGACTGGGACTTCTGG-3′) and Mb661R (5′-CCGGMGCAACGTCYTTACC-3′) [26], respectively. In a total volume of 20 μL, the qPCR amplifications were done using an SYBR@ (Takara Premix Ex Taq$^{TM}$, Dalian, China), with a reaction mixture containing 0.4 μL each for the forward and reverse primers (10 μmol L$^{-1}$), 10 μL ChamQ SYBR Color qPCR, 2 μL template DNA and 6.8 μL sterile water. Using different primers, each functional gene of the amplified fragments was cloned in a pMD 18-T vector and subsequently sequenced. Based on a 10-fold dilution of linear plasmid DNA, a triplicate of standard curves of all genes was prepared. Triplicate amplification was done expending 30 s at 95 °C, followed by 40 cycles

at 95 °C for 5 s, 60 °C for 34 s, and 72 °C for 15 s, and finally by dissociation at 95 °C for 15 s and 60 °C for 60 s [27].

### 2.6. Statistical Analysis

Data analyses were computed on SPSS version 23.0, IBM, Chicago, IL, USA. To test the differences between the treatments, analysis of variance (ANOVA) for randomized complete design was used while significant differences in treatments at $p < 0.05$ were performed using the Duncan multiple-range test. Microsoft Excel 2013 was used to plot graphs.

## 3. Results

### 3.1. CH$_4$ Emission

In the single cropping system, RedT + S produced significantly higher CH$_4$ flux than PT − S, RoT + S and PT + S on the 3rd, 4th and 5th week of gas sampling (Figure 2a). RedT + S recorded the highest significant CH$_4$ peak flux of 75.7 mg m$^{-2}$ h$^{-1}$ at the 5th week of gas sampling (Figure 2a). Compared to no straw incorporation (PT − S), straw incorporated tillage of RedT + S, RoT + S and PT + S increased cumulative CH$_4$ by 44.4% to 56.6% (Figure 3). Significantly higher cumulative CH$_4$ emission was produced by RedT + S compared to RoT + S and PT + S (Figure 3).

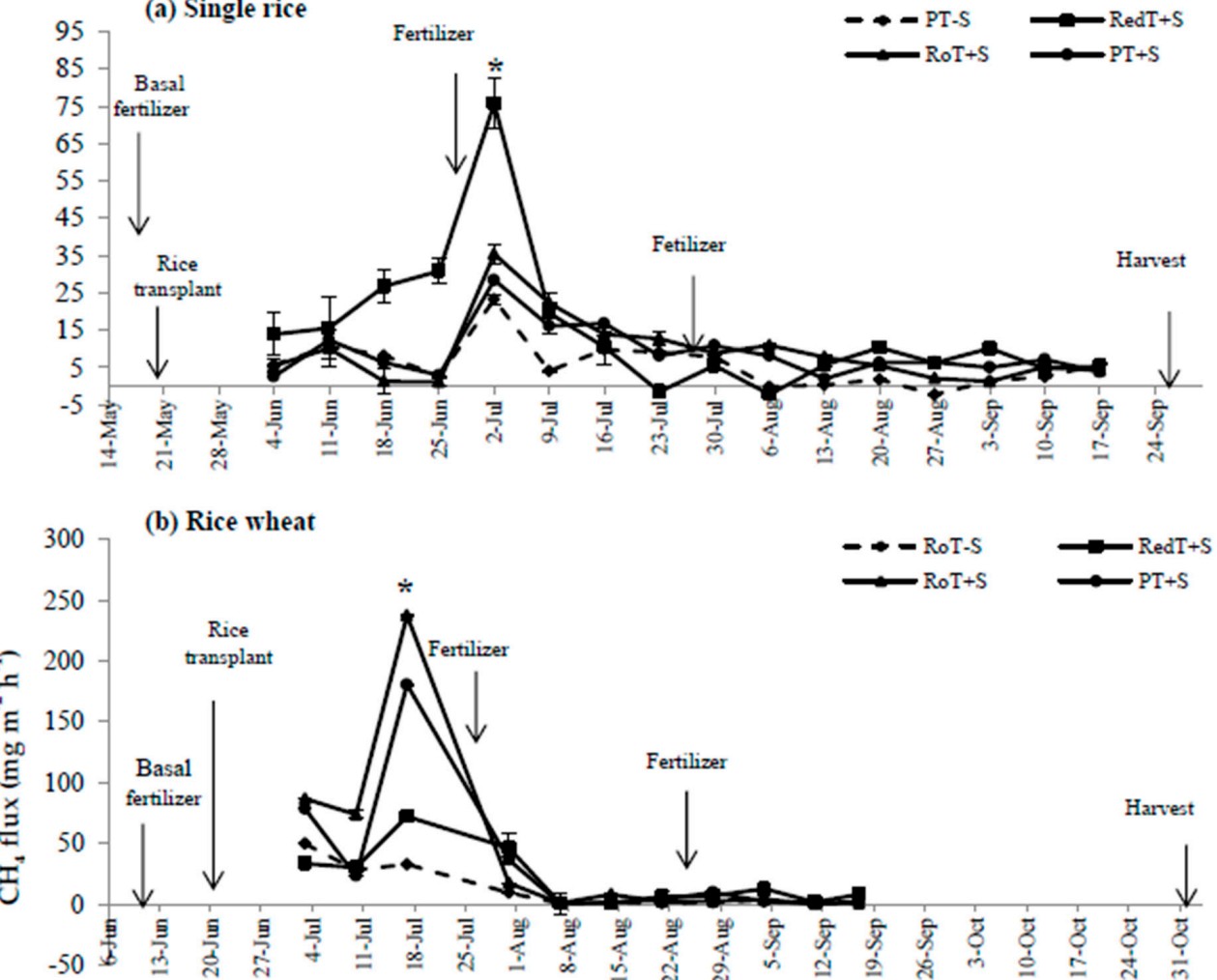

**Figure 2.** *Cont.*

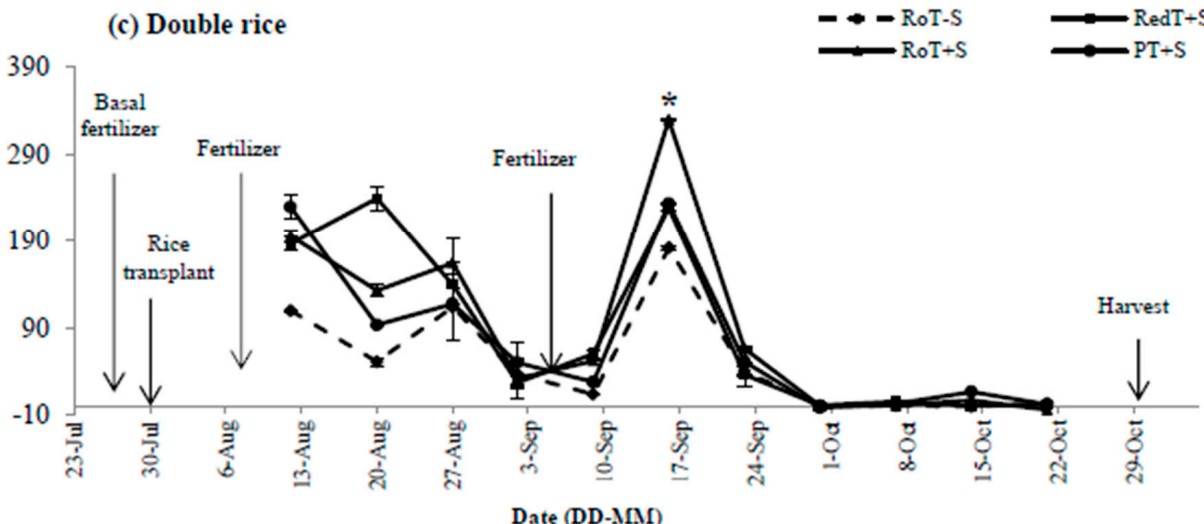

**Figure 2.** Impacts of straw incorporated tillage patterns on $CH_4$ fluxes in the three rice cropping systems. PT − S indicates plough tillage + no straw, RoT − S indicates rotary tillage + no straw, RedT + S indicates reduced tillage + straw, RoT + S indicates rotary tillage + straw; PT + S indicates plough tillage + straw. Since the baseline of soil tillage was different, PT − S and RoT − S were respectively implemented as the control in different test sites. * indicates a significant difference at 0.05 probability level in each cropping system.

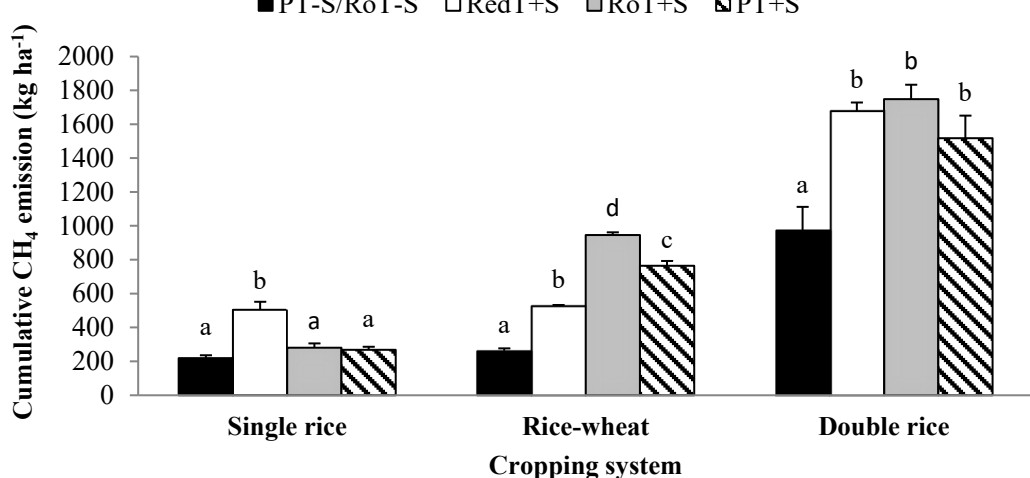

**Figure 3.** Impacts of straw incorporated tillage patterns on cumulative $CH_4$ emission in the three cropping systems. Different letters within the same column indicate significant differences among treatments at a 0.05 probability level in each cropping system.

In the rice-wheat cropping system, RoT + S recorded the highest and most significant $CH_4$ peak flux of 237.3 mg m$^{-2}$ h$^{-1}$ in comparison with 32.6 mg m$^{-2}$ h$^{-1}$ of RoT − S (Figure 2b). PT + S produced significantly 60.0% more $CH_4$ emissions and 39.1% fewer $CH_4$ emissions compared to RedT + S and RoT + S on 17th July respectively (Figure 2b). The straw incorporated tillage of RedT + S, RoT + S and PT + S significantly increased cumulative $CH_4$ emission compared to RoT − S (Figure 3). PT + S produced significantly 30.0% more cumulative $CH_4$ emission compared to RedT + S in the rice-wheat cropping system.

In the straw incorporated tillage methods of the double rice cropping system, PT + S, RoT + S and RedT + S produced 29.1% to 44.9 % more $CH_4$ emission compared to RoT − S (Figure 2c). Peak significant $CH_4$ flux of 329.2 mg m$^{-2}$ h$^{-1}$ was observed on the 6th week of rice growth in RoT + S while the lowest flux was noted in the RoT − S treatment (181.3 mg m$^{-2}$ h$^{-1}$) (Figure 2c). Significant cumulative $CH_4$ emission differences between

RoT − S and the straw incorporated tillage practices of RedT + S, RoT + S and PT + S were noted. The use of PT + S decreased cumulative $CH_4$ emission by 9.5% and 13.2% in comparison with RedT + S and RoT + S ($p > 0.05$, Figure 3).

### 3.2. N₂O Emission

In the single rice cropping system, significant peak $N_2O$ flux was noted on the 10th week of rice growth with RoT + S recording the highest of (0.20 mg m$^{-2}$ h$^{-1}$) while the lowest of (0.10 mg m$^{-2}$ h$^{-1}$) was observed adopting PT − S (Figure 4a). The highest and lowest cumulative $N_2O$ emission of 0.22 and 0.11 kg ha$^{-1}$ was observed in PT − S and RoT + S respectively in the single rice cropping system (Figure 5).

In the 8th week of rice growth, RoT + S and RedT + S recorded the highest and lowest $N_2O$ emission of 1.36 and 0.29 mg m$^{-2}$ h$^{-1}$ respectively in the rice-wheat cropping (Figure 4b). A 3.2 kg ha$^{-1}$ peak cumulative $N_2O$ emission was produced in the RoT + S treatment (Figure 5).

In the double rice cropping, peak positive and negative fluxes of $N_2O$ were noted in the 7th week of late rice growth (Figure 4c). The RoT + S produced the highest $N_2O$ flux of 0.61 mg m$^{-2}$ h$^{-1}$ while RoT − S recorded the lowest flux (−0.77 mg m$^{-2}$ h$^{-1}$). The highest cumulative $N_2O$ emission of 2.73 kg ha$^{-1}$ in the late rice season occurred in RoT + S (Figure 5).

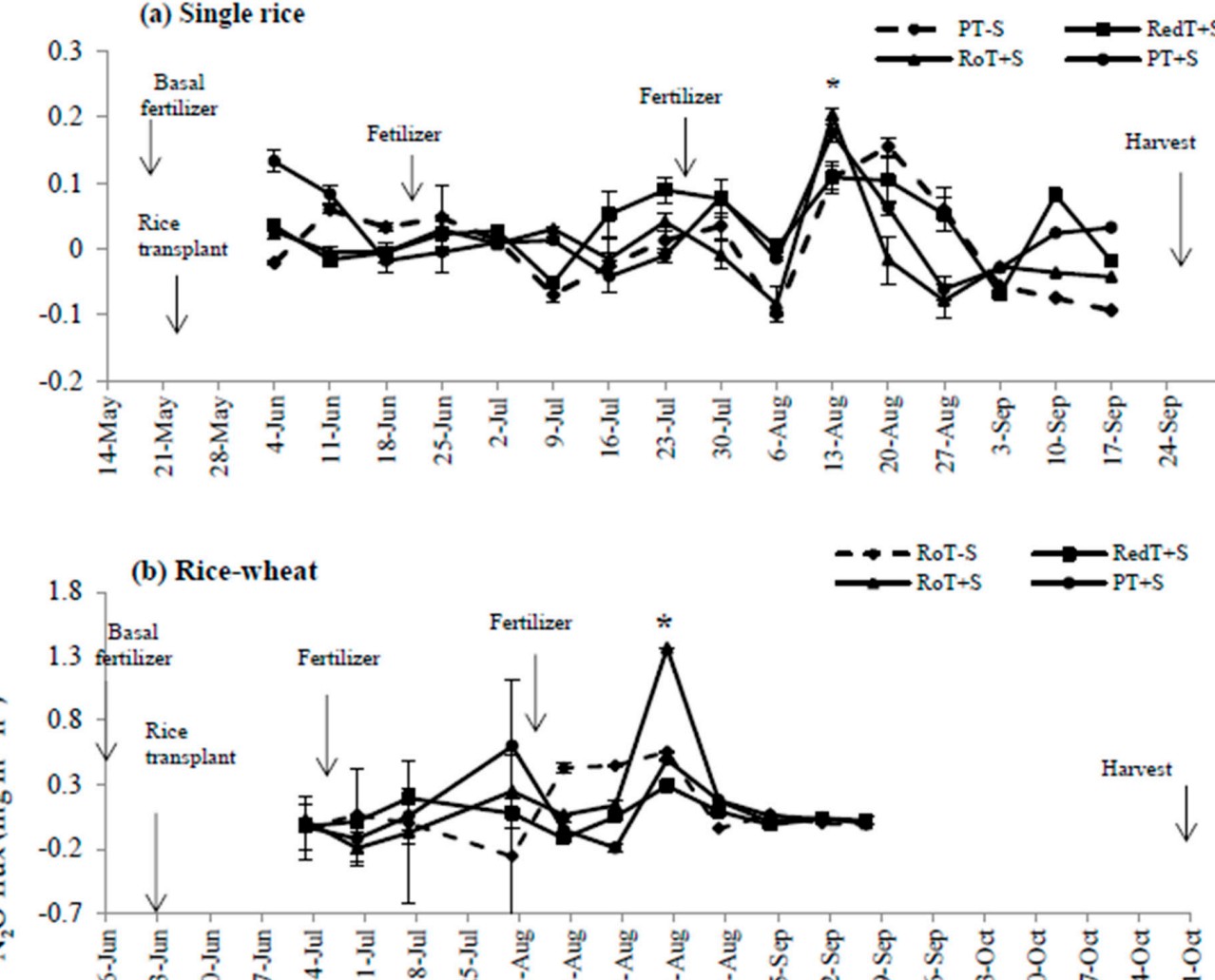

**Figure 4.** *Cont.*

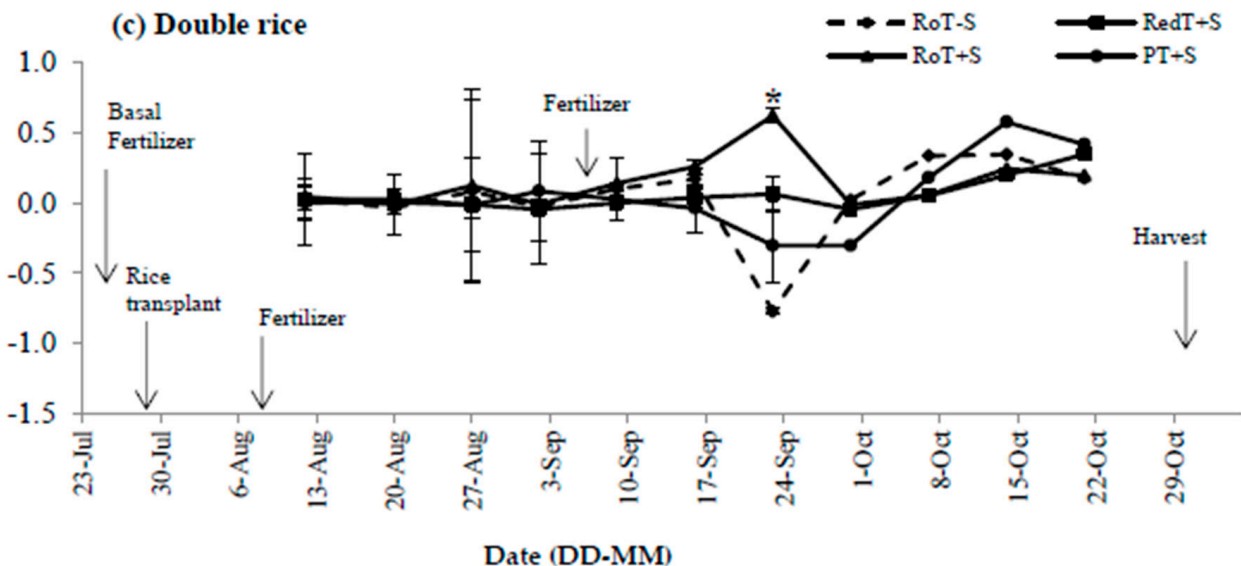

**Figure 4.** Impacts of straw incorporated tillage patterns on $N_2O$ fluxes in the three cropping systems. PT − S indicates plough tillage + no straw, RoT − S indicates rotary tillage + no straw, RedT + S indicates reduced tillage + straw, RoT + S indicates rotary tillage + straw; PT + S indicates plough tillage + straw. Since the baseline of soil tillage was different, PT − S and RoT − S were respectively implemented as the control in different test sites. * indicates a significant difference at 0.05 probability level in each cropping system.

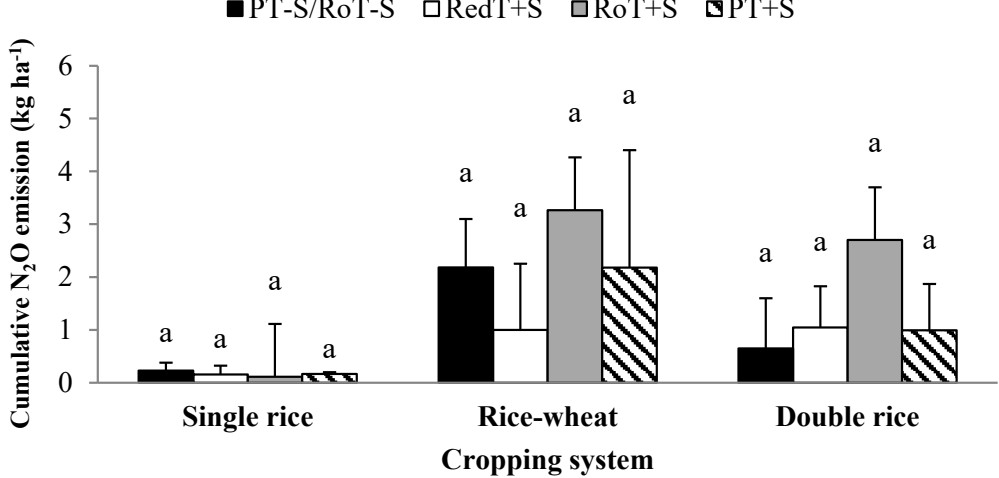

**Figure 5.** Impacts of straw incorporated tillage patterns on cumulative $N_2O$ emission Different letters within the same column indicate significant differences among treatments at 0.05 probability level in each cropping system.

### 3.3. Rice Yield

PT + S produced the highest rice yield of 9.1 t ha$^{-1}$ in the single rice cropping system. In the rice-wheat cropping system, the highest rice yield of 8.4 t ha$^{-1}$ occurred in RedT + S and RoT + S respectively and in the double rice cropping system, RedT + S recorded the highest late rice yield of 9.3 t ha$^{-1}$ (Table 2). Comparatively, both the single rice and double rice cropping systems produced significantly more rice compared to the rice-wheat cropping system (Table 2). There was significantly 9.3% more rice from the single rice cropping system and 11.3% more rice in the double rice cropping system than in the rice-wheat cropping system (Table 2).

**Table 2.** Impacts of straw incorporated tillage patterns on rice yield, area and yield-scaled emissions.

| Cropping System | Treatment | Yield (t ha$^{-1}$) | Area-Scaled Emission (t CO$_2$-eq ha$^{-1}$) | Yield-Scaled Emission (t CO$_2$-eq kg$^{-1}$) |
|---|---|---|---|---|
| Single rice | PT − S | 8.8 ± 0.13 a | 5.53 ± 0.40 a | 0.62 ± 0.01 a |
| | RedT + S | 8.2 ± 0.51 a | 12.64 ± 1.18 b | 1.53 ± 0.13 b |
| | RoT + S | 8.5 ± 0.47 a | 7.04 ± 0.62 a | 0.82 ± 0.09 a |
| | PT + S | 9.1 ± 0.21 a | 6.74 ± 0.45 a | 0.74 ± 0.06 a |
| | Mean | 8.6 ± 0.21 B | 8.69 ± 0.21 A | 0.93 ± 0.11 A |
| Rice-wheat | RoT − S | 7.7 ± 0.12 a | 7.49 ± 0.33 a | 0.96 ± 0.03 a |
| | RedT + S | 8.4 ± 0.81 a | 14.04 ± 0.40 b | 1.68 ± 0.13 b |
| | RoT + S | 7.0 ± 0.52 a | 25.57 ± 0.30 d | 3.68 ± 0.23 d |
| | PT + S | 8.1 ± 0.41 a | 20.52 ± 0.42 c | 2.55 ± 0.17 c |
| | Mean | 7.8 ± 0.27 A | 16.91 ± 2.05 B | 2.22 ± 0.31 B |
| Double rice | RoT − S | 9.1 ± 0.52 a | 24.54 ± 3.53 a | 2.66 ± 0.24 a |
| | RedT + S | 9.3 ± 0.50 a | 42.23 ± 1.51 b | 4.51 ± 0.09 b |
| | RoT + S | 8.2 ± 0.31 a | 44.51 ± 1.87 b | 5.38 ± 0.09 c |
| | PT + S | 8.4 ± 0.31 a | 38.22 ± 3.54 b | 4.54 ± 0.40 b |
| | Mean | 8.8 ± 0.23 B | 37.37 ± 2.61 C | 4.27 ± 0.32 C |

PT − S indicates plough tillage + no straw, RoT − S indicates rotary tillage + no straw, RedT + S indicates reduced tillage + straw, RoT + S indicates rotary tillage + straw; PT + S indicates plough tillage + straw. Since the baseline of soil tillage was different, PT − S and RoT − S were respectively implemented as the control in different test sites. Mean ± SE; different small letters within the same column indicate significance at the 0.05 probability level in each treatment. Different capital letters within the same column indicate significance at the 0.05 probability level in each cropping system.

*3.4. Area and Yield-Scaled GHG Emissions*

The RedT + S, RoT + S and PT + S of single rice cropping produced 44.3% to 56.2% more area-scaled emissions compared to the PT − S (Table 2). The RedT + S differed significantly from PT − S, RoT + S and PT + S in the single rice cropping system. PT + S decreased slightly area-scaled emissions by 4.1% compared to RoT + S. Similarly PT + S lowered significantly area-scaled emissions by 46.6% compared to RedT + S in the single rice cropping system (Table 2). Yield-scaled emission of RedT + S in the single rice cropping system was significantly different from PT − S, RoT + S and PT + S (Table 2). The PT + S reduced yield-scaled emission by 9.4% and 51.6% in comparison with RoT + S and RedT + S respectively.

Area-scaled emissions of RedT + S, RoT + S and PT + S of the rice-wheat cropping system were 19.7% to 70.7% significantly more compared to RoT − S (Table 2). RedT + S in terms of area-scaled emission was significantly lower than RoT + S and PT + S. The area-scaled emission of RoT + S and PT + S differed significantly from each other. The adoption of PT + S significantly reduced area-scaled emission by 19.7% in comparison to RoT + S (Table 2). In the rice-wheat cropping system, RedT + S recorded significantly 44.3% to 54.3% less yield-scaled emission compared with PT + S and RoT + S respectively (Table 2). The use of PT + S was significantly lower than RoT + S in terms of yield-scaled emission.

In the double rice cropping system the straw incorporated tillage methods produced significantly higher area-scaled emissions compared to RoT − S (Table 2). In comparison with the RoT − S, area-scaled emission increased by 36.8% to 47.6% in the straw incorporated tillage methods (Table 2). The use of PT + S decreased area-scaled emission by slightly by 9.5% and 14.1% compared to RedT + S and RoT + S respectively in the double rice cropping system. The RedT + S, RoT + S and PT + S of the double rice cropping system differed significantly from RoT − S and produced 41.0% to 50.5% more yield-scaled emission than RoT − S (Table 2). Yield-scaled emission of RoT + S varied significantly from PT + S and RoT + S (Table 2).

The area-scaled emission differed significantly among the three cropping systems with the single rice cropping system producing the lowest area-scaled emission (Table 2). The single rice cropping system differed significantly by 48.6% and 76.7% from the rice-wheat and double rice cropping systems respectively in terms of area-scaled emission. There was

a 54.7% significant difference in area-scaled emission between the rice-wheat and double rice cropping systems (Table 2). A comparison of the yield-scaled emission among the cropping systems showed that the double rice cropping system was 48.0% to 78.2% more significant than the rice-wheat and single rice cropping respectively (Table 2). Likewise, the rice-wheat cropping produced 56.7% more yield-scaled emission compared with the single rice cropping (Table 2). In the three cropping systems, yield-scaled emissions were in the order: single rice < rice-wheat < double rice cropping system (Table 2).

### 3.5. Impacts of Straw Incorporated Tillage Patterns on Methanogens and Methanotrophs

A comparison of methanogens in the 15 cm soil layer depth showed significant differences in the abundance of methanogens between RoT + S and PT − S in the single rice cropping system (Figure 6). For methanogenic abundance and methanotroph abundance, there is no significant difference among the PT + S, RoT + S and RedT + S (Figure 6).

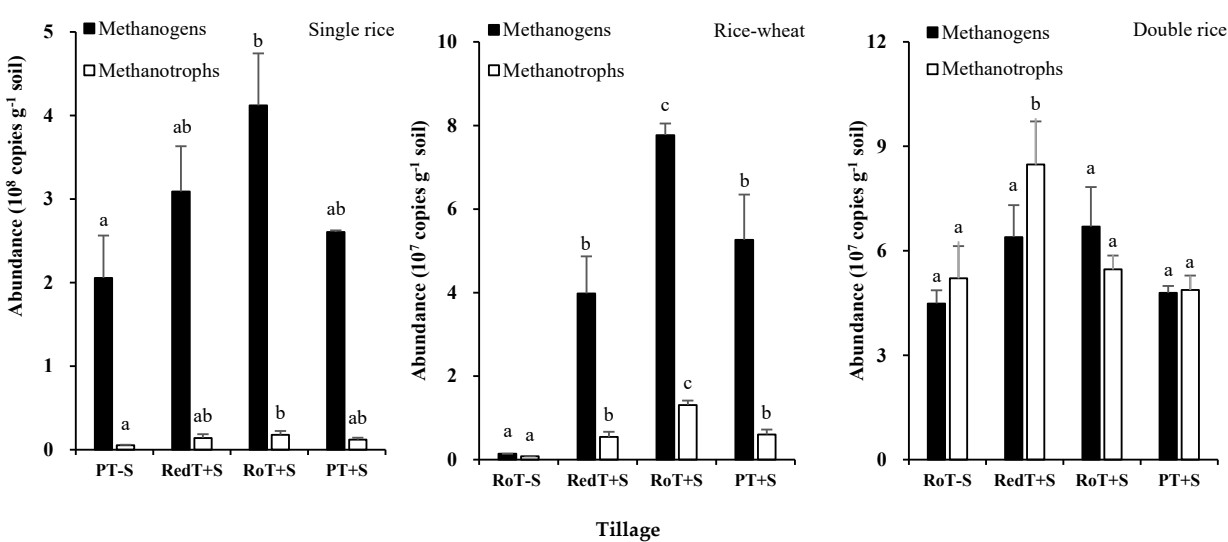

**Figure 6.** Impacts of straw incorporated tillage patterns on methanogens and methanotrophs PT − S indicates plough tillage + no straw, RoT − S indicates rotary tillage + no straw, RedT + S indicates reduced tillage + straw, RoT + S indicates rotary tillage + straw; PT + S indicates plough tillage + straw. Since the baseline of soil tillage was different, PT − S and RoT − S were respectively implemented as the control in different test sites. Mean ± SE; different small letters within the same column indicate significance at the 0.05 probability level in each treatment. Different capital letters within the same column indicate significance at the 0.05 probability level in each cropping system.

In the rice-wheat cropping system RedT + S, PT + S and RoT + S significantly increased the population of methanogens by more than 30.0% compared to RT − S (Figure 6). In comparison with RedT + S, the adoption of RoT + S recorded significantly 32.0% more methanogenic abundance while the methanogenic abundance of PT + S increased slightly (Figure 6). The RoT − S differed significantly in methanotroph abundance compared to RedT + S, PT + S and RoT + S. Methanotrophic abundance of RoT + S was 54.1% and 58.0% more and significantly different from PT + S and RedT + S respectively (Figure 6).

In the double rice cropping system, straw incorporation triggered a 3.0% to 38.5% increase in methanogenic abundance compared to RoT − S ($p < 0.05$) (Figure 6). The methanotrophs abundance in RedT + S was 35.4% and 42.4% more and significantly higher compared to RoT + S and PT + S respectively (Figure 6).

### 4. Discussion

Straw addition to reduced, rotary and plough tillage (Figures 2 and 3) was a precursor for higher $CH_4$ emission when it coincided with rice tillering [27,28] and was consistent with previous studies [5,29]. The RedT + S of single rice and RoT + S of rice-wheat and

double rice cropping systems respectively recorded the highest $CH_4$ emission (Figure 3), primarily due to the availability of straw as a labile organic substrate that degraded into $CH_4$ [30]. Though RedT + S has been shown to reduce SOC loss and GHG emissions owing to their minimal soil disturbance [31], higher emissions did occur under the single rice cropping system. We anticipated that slightly higher soil surface temperature was a trigger for an accelerated microbial breakdown of straw as compared to the colder soil temperature in the deeper soil depth of the single rice cropping system.

In the rice-wheat cropping system, RedT + S recorded the lowest $CH_4$ emission (Figure 3a, Table 2) compared to RoT + S and PT + S mainly owed to the reduction in soil water content via continuous dry-wet irrigation cycles adopted. It inhibited the formation of soil reductive conditions and $CH_4$ production, whiles positively affecting $CH_4$ oxidation by enhancing the soil aeration conditions [8,32]. This is recognized as a favorable preference for $CH_4$ mitigation [33].

In the double rice cropping system, there is no significant difference among the different tillage methods with straw incorporation mainly caused by the higher temperatures during rice tillering (Figure 1c). Higher temperatures offset the effects of different tillage methods on $CH_4$ emissions, resulting in no significant difference among methanogens [34].

Generally, plough tillage under straw incorporation had a positive effect on $CH_4$ emission mitigation in both the single rice and double rice cropping systems (Figure 3). It tended to reduce the abundance of methanogens at each location (Figure 6). This can be explained by the availability of straw substrate and the population size of methanogens [17,35]. Some studies have noted a positive correlation between cumulative $CH_4$ emission and methanogenic abundance [28,31]. In our study, a positive correlation between cumulative $CH_4$ emission and methanogenic abundance in the single rice ($r = 0.659$, $p < 0.05$), rice-wheat ($r = 0.899$, $p < 0.05$) and double rice cropping system ($r = 0.454$, $p > 0.05$) was noted (Figure S1).

The $N_2O$ flux and cumulative emission were generally low among the tillage practices of the single rice cropping system (Figures 4 and 5). Additionally, the contributions of $N_2O$ emissions to the area-scaled emission were low similar to previous studies [36]. This confirms the assertion by Jahangir et al. [37] that rice paddies are not sources of $N_2O$ emission.

Differences in the area and yield-scaled emissions (Table 2), in the paddy field, arose largely due to the $CH_4$ emission and yield differences among the tillage methods per previous studies [38]. In this study, with straw incorporation, higher $CH_4$ emission and lower rice yield of RedT + S in the single cropping system accounted for the highest yield-scaled emissions, while high rice yield in the straw incorporated plough and rotary tillage compensated for the higher cumulative $CH_4$ emissions. This subsequently lowered both the area and yield-scaled emissions (Table 2). As for the rice-wheat cropping system, the intermittent irrigation during the early tillering stage of RedT + S treatment lowered $CH_4$ emissions, resulting in lessening- area and yield-scaled emissions relative to RoT + S and PT + S. In the double rice cropping system, lower $CH_4$ emissions from PT + S and RedT + S minimized yield-scaled emissions relative to RoT + S (Table 2).

Cumulative $CH_4$ emissions arising from straw incorporated tillage in the late rice season of the double rice cropping system were 78.6% and 57.8% more in comparison with single rice cropping and rice-wheat cropping systems respectively. Likewise, the response of rice-wheat cropping to cumulative $CH_4$ emissions was 49.3% more compared to single rice cropping. According to Zhou et al. [39] a complex suite of factors such as soil properties, cropping system, water regime, fertilizers and climate may account for the observed differences. The adoption of the late rice season of double rice cropping has been noted to emit more GHG because it is characterized by two rice growing seasons [40]. In this study, the sensitivity of area-scaled emission response was 73.2% and 54.7% more significant in the late rice season of the double rice cropping system compared to the rice season of single rice and rice-wheat cropping system respectively. A 41.2% more area-scaled emission in the rice-wheat cropping system was noted compared to the single rice cropping system. Studies show that temperature increases generate considerable $CH_4$ emissions from rice paddies [41].

As noted, higher temperatures in the double cropping system triggered much more GHG emissions than the single rice as evidenced by a similar observation by Zhang et al. [38]. The single rice cropping region experiences colder temperatures compared with both rice-wheat and double rice cropping systems. The low temperatures inhibited the straw decomposition rate [42] and thus subsequently influenced the $CH_4$ emissions from the field. The late rice season of double rice cropping has been noted to be the hottest time of the summer [5], a precursor for an enhanced rate of straw decomposition and therefore higher GHG emission compared to the rice-wheat cropping system. As for the yield-scaled emissions, they varied significantly among the cropping systems (Table 2). Yield-scaled emissions from the double rice cropping system were 78.2% and 48.0% higher compared with single rice and rice-wheat cropping system respectively, while yield-scaled emission in the rice-wheat cropping system was 58.1% higher than single rice cropping, these trends among different cropping systems were consistent with the report of Feng et al. [22].

The effect of crop straw returning to the field largely depends on the tillage methods, and also its long-term effect on the growth of rice plants. Further research is needed on the long-term effects of different tillage methods in the different rice cropping systems. As the application area of straw returning becomes wider, some effects in the initial stage of straw returning still need more attention. For example, in the single rice cropping and double rice cropping regions, long-term rotary tillage and reduced/no-tillage tend to lead to a shallower soil plough layer and straw returning depth. This is not conducive to the improvement of rice field fertility and long-term sustainable development. The dry-wet alternate irrigation method in the rice-upland cropping system is indeed a relatively good measure for high-yield and emission-reduction in rice planting areas where irrigation and drainage are convenient, but in areas where the conditions are not met, plough tillage can be adopted because it increases the depth of straw burial appropriately and conducive for the improvement of paddy field fertility.

## 5. Conclusions

Straw incorporation influenced emissions from the rice fields irrespective of the tillage practice and cropping system adopted. In the three rice-based cropping systems, area and yield-scaled emissions were in the order single rice<, rice-wheat< and double rice cropping system. RoT + S and PT + S maintained rice high yield and lowered both area and yield-scaled emission in the single rice cropping system. The rice-wheat cropping system RedT + S sustained high rice yield and lowered area and yield-scaled emissions. Plough tillage (PT + S) of the double rice cropping systems decreased yield-scaled emissions while maintaining a high yield. Plough tillage with straw incorporation into deeper soil layers and less soil fragmentation will be a good approach to mitigate carbon emissions and improve soil fertility and sustainability of rice paddy fields.

**Supplementary Materials:** The following supporting information can be downloaded at: https://www.mdpi.com/article/10.3390/agronomy13030880/s1, Figure S1: Correlations between cumulative $CH_4$ emission and methanogenic.

**Author Contributions:** Conceptualization, W.Z.; methodology, W.Z. and J.Z.; software, F.D.; validation, W.Z., J.Z. and F.D.; formal analysis, F.D.; investigation, F.D., O.O.B., N.Z., W.D., K.Z. and C.L.; resources, G.L.; data curation, J.Z.; writing—original draft preparation, F.D.; writing—review and editing, F.D., O.O.B., Z.S. (Ziyin Shang),.W.Z., C.Z. and J.Z.; visualization, W.D. and J.Z.; supervision, W.D., K.Z., Z.S. (Zhenwei Song) and C.L.; project administration, A.D.; funding acquisition, W.Z. All authors have read and agreed to the published version of the manuscript.

**Funding:** This study was supported by the earmarked fund for Modern Agro-industry Technology Research System-Green manure (CARS-22), the Key Projects of Consultation and Evaluation of the Academic Department of the Chinese Academy of Sciences (2021-SM01-B-008), the Agricultural Science and Technology Innovation Project of the Chinese Academy of Agricultural Sciences (Y2021YJ02, CAAS-XTCX2016008) and the Jiangxi Key Research and Development Program (20171BBH80020).

**Data Availability Statement:** Not applicable.

**Conflicts of Interest:** The authors declare no conflict of interest.

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
