# Peer review of "Plough Tillage Maintains High Rice Yield and Lowers Greenhouse Gas Emissions under Straw Incorporation in Three Rice-Based Cropping Systems"

_agronomy, doi:10.3390/agronomy13030880_

Round 1

Reviewer 1 Report

Major comments:

Introduction: Needs to be rewritten. It is unclear from the introduction sections that the experiment compares different tillage methods with straw incorporation on GHG emissions. The effect of different tillage and straw incorporation on methane and nitrous oxide emissions is completely lacking.

Methods:

·         When and how were the soil samples collected for initial properties, and at what depth?

·         It is better to include the average air temperature and total precipitation in three cropping in table 1.

·         Include the timeline of different management practices. For e.g. when was it tilled, seeded, transplanted, fertilized, and harvested? Include the tillage and fertilization events in the methane and N2O emissions figures.

·         How frequently was gas sampled?

·         The chamber deployment time was only 15 minutes which may not be sensitive enough to capture the slight differences in N2O concentration. 

·         Was the gas not sampled after fertilization or just not a day of fertilization? If not sampled after the fertilization effect, the authors missed the window to capture the N2O emission peaks.

Results:

        Validate the results with statistics wherever relevant.

        If the differences are not significant, then avoid the comparison between the treatments.

Discussion:

The discussion is superficial; the effect of tillage and straw incorporation on methane emissions varies in three rice cropping systems. E.g. the emissions were higher in the reduced tillage with straw in single rice and the lowest in the rice-wheat cropping system. These differences should be discussed in detail with possible mechanisms. Also, it is noteworthy that there is no difference in methanogens or methanotrophs population among different straw addition treatments in double rice where the highest effect on methane emissions were observed.

The authors have discussed the positive effect of plough tillage and straw incorporation on methane emissions potential under both single-rice and double rice systems. However, according to the results in figure 5, there was no significant difference in methane emissions from RoT+S and PT+s in single rice and between any straw addition treatments in double –rice.

Also, the authors discussed the difference in N2O emissions among the treatments in the rice-wheat cropping system but the difference in seasonal variation was not significant, according to figure 6. 

Conclusion: The author has concluded that among the straw-incorporated tillage, plough tillage decreased the CH4 emissions in all cropping systems, which is not valid. In rice and rice systems, the CH4 emission from plough tillage and rotational tillage was similar. In the rice–wheat system, the reduced tillage has the lowest methane emissions; in double rice, there was no difference in the tillage system in terms of methane emissions.

Author Response

The comments from the Editor and reviewers are responded by points as follows:

Reviewer 1

Comments and Suggestions for Authors

Major comments:

  1. Introduction: Needs to be rewritten. It is unclear from the introduction sections that the experiment compares different tillage methods with straw incorporation on GHG emissions. The effect of different tillage and straw incorporation on methane and nitrous oxide emissions is completely lacking.

Reply: Thank you for your good suggestions. We have added one paragraph to compare different tillage methods with straw incorporation on methane and nitrous oxide emissions.

Line 46- Line 54: “Tillage effects on CH4 and N2O emission are not always consistent among different studies [5]. Conventional and rotary tillage increased N2O emissions by 2.3% to 30.8% and -2.3% to 47.4%, respectively [6]. Compared to plough tillage, no tillage decreased cumulative CH4 emissions by 10.1 % while rotary tillage increased CH4 emissions by 11.4 % [7]. Reduced tillage compared to conventional tillage minimized the global warming potential (GWP) of CH4 and N2O by 20.8% under the single crop system though not significant under the double crops rotation system [8]. Therefore, a detailed knowledge of the different tillage methods and straw incorporation in each cropping system on GHG emissions is imperative for the recommendation of low GHG emission practices.”

  1. Methods:

(1) When and how were the soil samples collected for initial properties, and at what depth?

Reply: Thank you. We have added the sampling details.

Line 275-line 276: “Three replicate of initial soil samples collected at a depth of 0 - 15cm per plot, were composited, air-dried and ground to pass through a 2mm sieve.”

(2) It is better to include the average air temperature and total precipitation in three cropping in table 1.

Reply: Thanks. We have added the average temperature and total precipitation in table 1.

Table 1. Weather conditions and initial soil properties

Cropping system

Average air temperature (0C)

Total precipitation (mm)

pH

Soil organic matter

(g kg-1)

Available N (mg kg-1)

Available P (mg kg-1)

Available K (mg kg-1)

Single rice

19.4

739.2

8.6

27.3

78.9

24.2

184.7

Rice-wheat

27.2

781.5

6.8

26.1

135.3

28.6

92.2

Double rice

25.9

350.2

5.9

22.6

128.5

57.2

100.3

(3) Include the timeline of different management practices. For e.g. when was it tilled, seeded, transplanted, fertilized, and harvested? Include the tillage and fertilization events in the methane and N2O emissions figures.

Reply: Thank you. We have added the timeline of tilled, seeded, transplanted, fertilized and harvested in Figure 2, and the tillage and fertilization events were marked the Figure 3 and Figure 5.

Figure 1. Daily average temperature, precipitation and management practices during rice growing seasons in the three cropping systems

Figure 2. Impacts of straw incorporated tillage patterns on CH4 fluxes in the three rice cropping systems.

Figure 4. Impacts of straw incorporated tillage patterns on N2O fluxes in the three rice cropping systems.

(3)  How frequently was gas sampled?

Reply: Thank you. The gas was sampled weekly. It has been inserted into section 2.3

Line 248-Line 251:” From 9:00 to 11:00 am weekly, the chamber headspace gas were sampled into pre-evacuated vacuum tubes via a 50 mL airtight syringe equipped with a three-way stop-cork at intervals of 0, 5, 10 and 15 minutes after chamber closure while an attached digital thermometer was used to read chamber temperature.”

(4) The chamber deployment time was only 15 minutes which may not be sensitive enough to capture the slight differences in N2O concentration.

Reply: Thanks for your good suggestions! As for chamber deployment time, it would be better if we lengthen the sampling time intervals appropriately, and 15 minutes is a common method to explore the CH4 and N2O flux in paddy field (Khalil and Rasmussen, 2002), and the determinations can prove present the greenhouse gas flux in the field. The N2O emissions in paddy field during rice season mostly is lower significantly than dryland crop seasons (eg. Wheat, oil seed, etc), and the proportions of total emission contributed by N2O emissions in rice season is less than 15% (Linquist et al., 2015; Wang et al., 2016). Actually, under the flooded conditions during rice growth season, N2O emission was not stable, leading to slight differences in N2O concentration.

Reference:

Khalil, M. A. K., Rasmussen, R. A. (2002). Seasonal Production and Emission of Methane from Rice Fields, Final Report. Portland State University, Portland, OR (US); Oregon Graduate Institute, Beaverton, OR (US). DE-FG03-97ER62401.

Linquist BA, Anders MM, Adviento-Borbe MA, Chaney RL, Nalley LL, da Rosa EF, van Kessel C (2015) Reducing greenhouse gas emissions, water use, and grain arsenic levels in rice systems. Glob Change Biol 21:407–417.

Wang W, Wu X, Chen A, Xie X, Wang Y, Yin C (2016) Mitigating effects of ex situ application of rice straw on CH4 and N2O emissions from paddy-upland coexisting system. Sci Rep 6:37402.

(5) Was the gas not sampled after fertilization or just not a day of fertilization? If not sampled after the fertilization effect, the authors missed the window to capture the N2O emission peaks.

Reply: Thank you. The proportions of total emission contributed by N2O emissions in rice season is less than 15% (Wang et al., 2016; Linquist et al., 2015). When we applied nitrogen fertilizer, the water depth in the field was around 5 cm depth, so under the flooded conditions, there was less N2O emission. Also, to avoid the fertilizer chasing effects, greenhouse gas sampling date were chosen not to coincide with the date of fertilizer application.

  1. Results:
  • Validate the results with statistics wherever relevant.

Reply: Thank you. We have corrected the descriptions with statistics.

Line 312 - Line 314: “RedT + S produced significantly higher CH4 flux than PT - S, RoT + S and PT + S from the 3rd to 5th week of gas collection. RedT + S produced the highest significant CH4 flux of 75.7 mg m-2 h-1than RoT”

Line 317 -Line 319: “). Significantly higher cumulative CH4 emission was produced by RedT + S compared to RoT + S and PT + S (Figure 4).”

Line 320 -Line 326: “In the rice-wheat cropping system, RoT + S recorded the highest and significant CH4 flux of 237.3 mg m-2 h-1 in comparison with 32.6 mg m-2 h-1 produced by RoT - S (Figure 3b). PT + S produced significantly 60.0% more CH4 emission and 39.1% less CH4 emis-sions compared to RedT + S and RoT + S respectively (Figure 3b)”

Line 470 - Line 477  “The RedT + S differed significantly from PT – S, RoT + S and PT + S in the single rice cropping system. PT + S decreased area-scaled emissions by 4.1% compared to RoT + S. Similarly PT + S lowered area-scaled emissions by 46.6% compared to RedT + S in the single rice”

Line 480 - Line 485: “area-scaled emission was significantly lower than RoT + S and PT + S.  The area-scaled emission of RoT + S and PT + S also differed significantly. The adoption of PT + S reduced area-scaled emission by 19.7% in comparison to RoT + S (Table 2). In the rice-wheat cropping system”

Line 491 - Line 506: “The RedT + S, RoT + S and PT + S of the double rice cropping system differed significantly from RoT – S and produced 41.0% to 50.5% more yield-scaled emission than RoT - S (Table 2). Yield-scaled emission of RoT + S varied significantly from PT + S and RoT + S (Table 2)”

Line 540 - Line 557: “A comparison of methanogens in the 15cm soil layer depth showed significant differences in the abundance of methanogens between RoT + S and PT – S in the single rice cropping system (Figure 7).”

  • If the differences are not significant, then avoid the comparison between the treatments.

Reply: Thank you. We have deleted the comparison between treatments with no significant differences.

  1. Discussion:
  • The discussion is superficial; the effect of tillage and straw incorporation on methane emissions varies in three rice cropping systems. E.g. the emissions were higher in the reduced tillage with straw in single rice and the lowest in the rice-wheat cropping system. These differences should be discussed in detail with possible mechanisms. Also, it is noteworthy that there is no difference in methanogens or methanotrophs population among different straw addition treatments in double rice where the highest effect on methane emissions were observed.

Reply: Thanks for your good suggestions!

Lines 586-614: “Straw addition to reduced, rotary and plough tillage (Figure 2,3) was a precursor for higher CH4 emission when it coincided with rice tillering [27,28] and was consistent with previous studies [5,29]. The RedT + S of single rice and RoT + S of rice-wheat and double rice cropping system respectively recorded the highest CH4 emission (Figure 3), primarily due to the availability of straw as a labile organic substrate that degraded into CH4 [30]. Though RedT + S has been shown to reduce SOC loss and GHG emissions owning to their minimal soil disturbance [31], higher emission did occur under the single rice cropping system. We anticipated that slightly higher soil surface temperature was a trigger for accelerated microbial breakdown of straw as compared to the colder soil temperature in deeper soil depth of the single rice cropping system. In the rice-wheat cropping system, RedT + S recorded the lowest CH4 emission (Figure 3a, Table 2) compared to RoT + S and PT + S mainly owed to the reduction in soil water content via continuous dry-wet irrigation cycles adopted. It inhibited the formation of soil reductive conditions and CH4 production, whiles positively affecting CH4 oxidation by enhancing the soil aeration conditions [32,33]. This is recognized as a favorable preference for CH4 mitigation [34]. In double rice cropping system, there is no significant difference among the different tillage methods with straw incorporation mainly caused by the higher temperatures during rice tillering (Figure 1c). Higher temperatures offset the effects of different tillage methods on CH4 emissions, resulting in no significant difference among of methanogens [35]”

  • The authors have discussed the positive effect of plough tillage and straw incorporation on methane emissions potential under both single-rice and double rice systems. However, according to the results in figure 5, there was no significant difference in methane emissions from RoT+S and PT+s in single rice and between any straw addition treatments in double –rice.

Reply: Thanks for your good suggestions!

In single rice cropping system, both ROT+S and PT+S decreased significantly the cumulative CH4 emission, area- and yield- scared GHGs emission when compared to RedT+S, however, there is no significant difference between ROT+S and PT+S. In double rice cropping system, though there is no significant difference of cumulative CH4 emission and area-scared GHGs emission, the yield-scared emission of RedT+S and PT+S were significantly lower than RoT +S. However, there is no significant difference between RedT+S and PT+S. Considering comprehensively the soil fertility improvement and sustainability of paddy field, plough tillage with straw incorporation into deeper soil layers and less soil fragmentation will be a good approach to mitigate carbon emissions while maintaining high rice yield”.

Line 657 - Line 669, Discussion section: The effect of crop straw returning to the field largely depends on the tillage methods, and also its long-term effect on the growth of rice plants. Further research is needed on the long-term effects of different tillage methods in the different rice crop-ping systems. As the application area of straw returning becomes wider, some effects in the initial stage of straw returning still need more attention. For example, in the single rice cropping and double rice cropping regions, long-term rotary tillage and reduced/no-tillage tend to lead to a shallower soil plough layer and straw returning depth. This is not conducive for the improvement of rice field fertility and long-term sustain-able development. For the dry-wet alternate irrigation method in the rice- upland cropping system, this is indeed a relatively good measure for high-yield and emission-reduction in rice planting areas where irrigation and drainage are convenient, but in areas where the conditions are not met, plough tillage can be adopted because it in-creases the depth of straw burial appropriately and conducive for the improvement of paddy field fertility.”

Line 689- Line 697, Conclusion: “In the three rice based cropping systems, area and yield-scaled emissions was in the order single rice<, rice-wheat< and double rice cropping system. RoT + S and PT + S maintained rice high yield and lowered both area and yield-scaled emission in single rice cropping system. The rice-wheat cropping system ReDT + S sustained high rice yield and lowered area and yield-scaled emissions. Plough tillage (PT + S) of the double rice cropping systems decreased yield-scaled emissions while maintaining high yield. Plough tillage with straw incorporation into deeper soil layers and less soil fragmentation will be a good approach to mitigate carbon emissions, improve soil fertility and sustainability of rice paddy fields.”

  • Also, the authors discussed the difference in N2O emissions among the treatments in the rice-wheat cropping system but the difference in seasonal variation was not significant, according to figure 6.

Reply: Thanks for your good suggestions! The sentence has been deleted

  1. Conclusion: The author has concluded that among the straw-incorporated tillage, plough tillage decreased the CH4 emissions in all cropping systems, which is not valid. In rice and rice systems, the CH4 emission from plough tillage and rotational tillage was similar. In the rice–wheat system, the reduced tillage has the lowest methane emissions; in double rice, there was no difference in the tillage system in terms of methane emissions.

Reply: Thank you. We have rewritten the conclusion.

Line 688- Line 697: “Straw incorporation influenced emissions from the rice fields irrespective of the tillage practice and cropping system adopted. In the three rice based cropping systems, area and yield-scaled emissions was in the order single rice<, rice-wheat< and double rice cropping system. RoT + S and PT + S maintained rice high yield and lowered both area and yield-scaled emission in single rice cropping system. The rice-wheat cropping system ReDT + S sustained high rice yield and lowered area and yield-scaled emissions. Plough tillage (PT + S) of the double rice cropping systems decreased yield-scaled emissions while maintaining high yield. Plough tillage with straw incorporation into deeper soil layers and less soil fragmentation will be a good approach to mitigate carbon emissions, improve soil fertility and sustainability of rice paddy fields”

Reviewer 2 Report

This manuscript addresses an important question around trade-offs between using crop residue to improve soil health and the potential for increased emission of GHG. The study is well designed with appropriate methodology and statistical models. However, the presentation of results and associated conclusions are not well presented. For example, there are several instances where the author documents differences when in fact there are no differences, at least statistically. There are also several instances where results or explanations are vague and not well presented, leaving the reader to interpret statements and conclusions.  Below are a few suggested revisions/questions. This is not an exhaustive list as these same issues kept coming up throughout the paper. Overall, this is a good experiment - the authors just need to be much more detailed and focused when presenting results and conclusions. 

Line 290-293: This section appears to only be applicable to the 2-Jul date, yet there is no reference to date.

Line 294: The only straw incorporated tillage treatment that increased cumulative CH4 was RedT+S – the others are not significantly different from PT-S, so why the range?

Line 295: Figure 5 is mentioned before Figure 4 – figures need to be numbered in the order they are mentioned in the text.

Line 295: Suggest revising to “The adoption of RedT+S resulted in significantly higher cumulative CH4 emissions than all other tillage treatments”. Need more detail when writing statements like this…

Line 296: PT+S was not significantly different from RoT+S and RedT+S, so how can you state that it was 4.3 and 46.8% less?

L297: Start new paragraph when transitioning to new cropping systems, i.e., rice-wheat.

L297-299: Again, this seems to relate to only one time point. Please make that point.

L302: Reference Figure 5 when discussing cumulative CH4.

L303: Remove ‘however’

L309: You state that there are not significant differences amid straw incorporated tillage, yet you go on to talk about differences. If there are no differences, then there are no differences.

L313: Figure 5 shows no significant difference among treatments where straw was incorporated, yet you go on to talk about differences… Please don’t do that!

L360 – 375: This is well written. Follow this template when presenting results of CH4 emission.

L425: Reference Table 2

L423: I don’t understand why the authors are mentioning differences and then go on to say there are no differences…

L431-433: Again, the authors talk about differences that aren’t based on statistics in the first sentence but in the second sentence indicate that there are no differences. The second sentence is the only statement that can be made.

L555-560: The first sentence of this paragraph indicates that the incorporation of straw showed varied responses to GHG emissions. Yet, the next two sentences do not mention anything about straw incorporation, focused instead of average CH4 emissions between cropping systems. If straw incorporation mad a difference, which it did, than why talk about averages?

592: …”contributed significantly reducing cumulative CH4 emission” – compared to what?

595: RedT+S recorded the lowest CH4 emission? What figure are you referring to? If Figure 5, then this is not true – PT-S/RoT-S recorded the lowest emission; unless you are only referring to differences among treatments that included straw incorporation. More detail is needed throughout the paper.

L640-642: Among straw incorporated tillage, plough tillage decreased CH4 emissions across all three cropping systems? No. Each cropping system was different with respect to CH4 emission among tillage treatments.

Author Response

Reviewer 2

Comments and Suggestions for Authors

This manuscript addresses an important question around trade-offs between using crop residue to improve soil health and the potential for increased emission of GHG. The study is well designed with appropriate methodology and statistical models. However, the presentation of results and associated conclusions are not well presented. For example, there are several instances where the author documents differences when in fact there are no differences, at least statistically. There are also several instances where results or explanations are vague and not well presented, leaving the reader to interpret statements and conclusions.  Below are a few suggested revisions/questions. This is not an exhaustive list as these same issues kept coming up throughout the paper. Overall, this is a good experiment - the authors just need to be much more detailed and focused when presenting results and conclusions. 

Reply: Thank you very much for your helpful suggestions. We have double checked whole text and revised the results, discussion and conclusion sections accordingly.

  1. Line 290-293: This section appears to only be applicable to the 2-Jul date, yet there is no reference to date.

Reply: Thank you! We have revised the description to reflect the emission observed.

Line 307 – Line 313: “In the single cropping system, RedT + S produced significantly higher CH4 flux than PT - S, RoT + S and PT + S on the 3rd, 4th and 5th week of gas sampling (Figure 2a). RedT + S recorded the highest significant CH4 peak flux of 75.7 mg m-2 h-1 at the 5th week of gas sampling (Figure 2a).

  1. Line 294: The only straw incorporated tillage treatment that increased cumulative CH4 was RedT+S – the others are not significantly different from PT - S, so why the range?

Reply: Thank you! We have revised the description to the differences between treatments.

Line 310 – Lines 313: “Compared to no straw incorporation (PT – S), straw incorporated tillage of RedT + S, RoT + S and PT + S increased cumulative CH4 by 44.4% to 56.6% (Figure 3). Significantly higher cumulative CH4 emission was produced by RedT + S compared to RoT + S and PT + S (Figure 3). ”

  1. Line 295: Figure 5 is mentioned before Figure 4 – figures need to be numbered in the order they are mentioned in the text.

Reply: Thank you! The numbering of the figures has been changed.

  1. Line 295: Suggest revising to “The adoption of RedT+S resulted in significantly higher cumulative CH4 emissions than all other tillage treatments”. Need more detail when writing statements like this…

Reply: Thank you! We have revised the comparisons between treatments.

Line 312 – Line 313: “Significantly higher cumulative CH4 emission was produced by RedT + S compared to RoT + S and PT + S (Figure 3)”

  1. Line 296: PT+S was not significantly different from RoT+S and RedT+S, so how can you state that it was 4.3 and 46.8% less?

Reply: Thank you! We have revised the description of the differences between treatments.

Line 314 – Line 320: “In the rice-wheat cropping system, RoT + S recorded the highest and significant CH4 peak flux of 237.3 mg m-2 h-1 in comparison with 32.6 mg m-2 h-1 of RoT - S (Figure 2b). PT + S produced significantly 60.0% more CH4 emission and 39.1% less CH4 emissions compared to RedT + S and RoT + S on 17th July respectively (Figure 2b). The straw in-corporated tillage of RedT + S, RoT + S and PT + S significantly increased cumulative CH4 emission compared to RoT – S (Figure 3). PT + S produced significantly 30.0% more cumulative CH4 emission compared to RedT + S in the rice-wheat cropping system.”

  1. L297: Start new paragraph when transitioning to new cropping systems, i.e., rice-wheat.

Reply: Thank you! The comment has been well noted and corrected in the manuscript.

Line 314- Line 320: “In the rice-wheat cropping system, RoT + S recorded the highest and significant CH4 peak flux of 237.3 mg m-2 h-1 in comparison with 32.6 mg m-2 h-1 of RoT - S (Figure 2b). PT + S produced significantly 60.0% more CH4 emission and 39.1% less CH4 emissions compared to RedT + S and RoT + S on 17th July respectively (Figure 2b). The straw in-corporated tillage of RedT + S, RoT + S and PT + S significantly increased cumulative CH4 emission compared to RoT – S (Figure 3). PT + S produced significantly 30.0% more cumulative CH4 emission compared to RedT + S in the rice-wheat cropping system.

 L297-299: Again, this seems to relate to only one time point. Please make that point.

Reply: Thank you! We have revised the description to the differences between treatments.

Line 320 – Line 326: “In the rice-wheat cropping system, RoT + S recorded the highest and significant CH4 peak flux of 237.3 mg m-2 h-1 in comparison with 32.6 mg m-2 h-1 of RoT - S (Figure 2b). PT + S produced significantly 60.0% more CH4 emission and 39.1% less CH4 emissions compared to RedT + S and RoT + S on 17th July respectively (Figure 2b). The straw incorporated tillage of RedT + S, RoT + S and PT + S significantly increased cumulative CH4 emission compared to RoT – S (Figure 3). PT + S produced significantly 30.0% more cumulative CH4 emission compared to RedT + S in the rice-wheat cropping system.  

L302: Reference Figure 5 when discussing cumulative CH4.

Reply: Thank you! Reference to the appropriate figure has been made in the discussion of cumulative CH4.

Line 595 – Line 599: “Straw addition to reduced, rotary and plough tillage (Figure 2,3) was a precursor for higher CH4 emission when it coincided with rice tillering [27,28] and was consistent with previous studies [5,29]. The RedT + S of single rice and RoT + S of rice-wheat and double rice cropping system respectively recorded the highest CH4 emission (Figure 3), primarily due to the availability of straw as a labile organic substrate that degraded into CH4 [30].

Line 605 – Line 607: “In the rice-wheat cropping system, RedT + S recorded the lowest CH4 emission (Figure 3a, Table 2) compared to RoT + S and PT + S mainly owed to the reduction in soil water content via continuous dry-wet irrigation cycles adopted”.

  1. L303: Remove ‘however’

Reply: Thank you! “however” has been removed from the sentence.

  1. L309: You state that there are not significant differences amid straw incorporated tillage, yet you go on to talk about differences. If there are no differences, then there are no differences.

Reply: Thank you! The sentence has been deleted from the manuscript.

  1. L313: Figure 5 shows no significant difference among treatments where straw was incorporated, yet you go on to talk about differences… Please don’t do that!

Reply: Thank you! The sentence has been deleted.

  1. L360 – 375: This is well written. Follow this template when presenting results of CH4 emission.

Reply: Thank you! Comment well noted and effected in the manuscript.

  1. L425: Reference Table 2

Reply: Thank you! Reference has been made to Table 2.

  1. L423: I don’t understand why the authors are mentioning differences and then go on to say there are no differences…

Reply: Thank you! The sentence has been reframed to remove all ambiguities.

Line 456 – Line 463: “PT + S produced the highest rice yield of 9.1 t ha-1 in the single rice cropping system. In rice-wheat cropping system, the highest rice yield of 8.4 t ha-1occurred in RedT + S and RoT + S respectively and in the double rice cropping system, RedT + S recorded the highest late rice yield of 9.3 t ha-1 (Table 2). Comparatively both the single rice and double rice cropping systems produced significantly more rice compared to the rice-wheat cropping system (Table 2). There was significantly 9.3% more rice from the single rice cropping system and 11.3% more rice in the double rice cropping system than the rice-wheat cropping system (Table 2).

  1. L431-433: Again, the authors talk about differences that aren’t based on statistics in the first sentence but in the second sentence indicate that there are no differences. The second sentence is the only statement that can be made.

Reply: Thank you! We have revised the description to the differences between treatments.

Line 465 – Line 503:  “The RedT + S, RoT + S and PT + S of single rice cropping produced 44.3% to 56.2 % more area-scaled emissions compared to the PT – S (Table 2). The RedT + S differed significantly from PT – S, RoT + S and PT + S in the single rice cropping system. PT + S de-creased slightly area-scaled emissions by 4.1% compared to RoT + S. Similarly PT + S lowered significantly area-scaled emissions by 46.6% compared to RedT + S in the single rice cropping system (Table 2). Yield-scaled emission of RedT + S in the single rice crop-ping system was significantly different from PT - S, RoT + S and PT + S (Table 2). The PT + S reduced yield-scaled emission by 9.4% and 51.6% in comparison with RoT + S and RedT + S respectively.

Area-scaled emissions of RedT + S, RoT + S and PT + S of the rice-wheat cropping system were 19.7% to 70.7% significantly more compared to RoT – S (Table 2). RedT + S in terms of area-scaled emission was significantly lower than RoT + S and PT + S.  The ar-ea-scaled emission of RoT + S and PT + S differed significantly from each other. The adoption of PT + S significantly reduced area-scaled emission by 19.7% in comparison to RoT + S (Table 2). In the rice-wheat cropping system, RedT + S recorded significantly 44.3% to 54.3% less yield-scaled emission compared with PT + S and RoT + S respectively (Table 2). The use of PT + S was significantly lower than RoT + S in terms of yield-scaled emission.

In the double rice cropping system the straw incorporated tillage methods produced significantly higher area-scaled emission compared to RoT - S (Table 2). In com-parison with the RoT - S, area-scaled emission increased by 36.8% to 47.6% in the straw incorporated tillage methods (Table 2). The use of PT + S decreased area-scaled emission by slightly by 9.5% and 14.1% compared to RedT + S and RoT +S respectively in the double rice cropping system. The RedT + S, RoT + S and PT + S of the double rice crop-ping system differed significantly from RoT – S and produced 41.0% to 50.5% more yield-scaled emission than RoT - S (Table 2). Yield-scaled emission of RoT + S varied significantly from PT + S and RoT + S (Table 2). 

The area-scaled emission differed significantly among the three cropping systems with the single rice cropping system producing the lowest area-scaled emission (Table 2). The single rice cropping system differed significantly by 48.6% and 76.7% from the rice-wheat and double rice cropping systems respectively in terms of area-scaled emission. There was a 54.7% significant difference in area-scaled emission between the rice-wheat and double rice cropping systems (Table 2). A comparison of the yield-scaled emission among the cropping systems showed that the double rice cropping system was 48.0% to 78.2% more significant than the rice-wheat and single rice cropping respectively (Table 2). Likewise the rice-wheat cropping produced 56.7% more yield-scaled emission compared with the single rice cropping (Table 2). In the three cropping systems, yield-scaled emissions was in the order single rice < rice-wheat< double rice cropping system (Table 2).

  1. L555-560: The first sentence of this paragraph indicates that the incorporation of straw showed varied responses to GHG emissions. Yet, the next two sentences do not mention anything about straw incorporation, focused instead of average CH4 emissions between cropping systems. If straw incorporation made a difference, which it did, than why talk about averages?

Reply: Thank you for the good suggestion. The paragraphs have been re-written to reflect the corrections suggested.

Lines 599- Line 619: “Straw addition to reduced, rotary and plough tillage (Figure 2,3) was a precursor for higher CH4 emission when it coincided with rice tillering [27,28] and was consistent with previous studies [5,29]. The RedT + S of single rice and RoT + S of rice-wheat and double rice cropping system respectively recorded the highest CH4 emission (Figure 3), primarily due to the availability of straw as a labile organic substrate that degraded into CH4 [30]. Though RedT + S has been shown to reduce SOC loss and GHG emissions owning to their minimal soil disturbance [31], higher emission did occur under the single rice cropping system. We anticipated that slightly higher soil surface temperature was a trigger for accelerated microbial breakdown of straw as compared to the colder soil temperature in deeper soil depth of the single rice cropping system”

In the rice-wheat cropping system, RedT + S recorded the lowest CH4 emission (Figure 3a, Table 2) compared to RoT + S and PT + S mainly owed to the reduction in soil water content via continuous dry-wet irrigation cycles adopted. It inhibited the formation of soil reductive conditions and CH4 production, whiles positively affecting CH4 oxidation by enhancing the soil aeration conditions [32,33]. This is recognized as a favorable preference for CH4 mitigation [34].

In double rice cropping system, there is no significant difference among the dif-ferent tillage methods with straw incorporation mainly caused by the higher tempera-tures during rice tillering (Figure 1c). Higher temperatures offset the effects of different tillage methods on CH4 emissions, resulting in no significant difference among of methanogens [35].

  1. L592: …”contributed significantly reducing cumulative CH4 emission” – compared to what?

Reply: Thank you! “compared to RedT + S” has been inserted into the sentence.

Line 609-Line 610: “In the single rice cropping, the PT + S and RoT + S contributed significantly in reducing cumulative CH4 emission compared to RedT + S (Figure 3),”

  1. L595: RedT+S recorded the lowest CH4 emission? What figure are you referring to? If Figure 5, then this is not true – PT-S/RoT-S recorded the lowest emission; unless you are only referring to differences among treatments that included straw incorporation. More detail is needed throughout the paper.

Reply: Thank you!  The above sentence referred to Figure 4 and has subsequently been inserted.

Line 617 - Line 618: “In the single rice cropping, the PT + S and RoT + S contributed significantly in reducing cumulative CH4 emission compared to RedT + S (Figure 3)”

  1. L640-642: Among straw incorporated tillage, plough tillage decreased CH4 emissions across all three cropping systems? No. Each cropping system was different with respect to CH4 emission among tillage treatments.

Reply: Thank you!  The cropping system differences has been highlighted in the manuscript

 Line 672 - Line 684: “The effect of crop straw returning to the field largely depends on the tillage methods, and also its long-term effect on the growth of rice plants. Further research is needed on the long-term effects of different tillage methods in the different rice cropping systems. As the application area of straw returning becomes wider, some effects in the initial stage of straw returning still need more attention. For example, in the single rice cropping and double rice cropping regions, long-term rotary tillage and reduced/no-tillage tend to lead to a shallower soil plough layer and straw returning depth. This is not conducive for the improvement of rice field fertility and long-term sustainable development. For the dry-wet alternate irrigation method in the rice- upland cropping system, this is indeed a relatively good measure for high-yield and emission-reduction in rice planting areas where irrigation and drainage are convenient, but in areas where the conditions are not met, plough tillage can be adopted because it increases the depth of straw burial appropriately and conducive for the improvement of paddy field fertility.”

Reviewer 3 Report

The methodology appears well developed and described, as well as the use of statistics is appropriate and accurate. References are rich and appropriate. The manuscript is clear and well structured. English language is correct and fluent.

The paper is scientifically sound, and the experimental design appears generally appropriate.

Figures/tables/images/schemes are appropriate and properly show the data. They are easy to be interpreted and understood, even if some of them could be re-edited in order to enhance legibility (see specific comments).

Data is interpreted appropriately and consistently throughout the manuscript. Conclusions are consistent with the evidence and arguments presented.

Specific comments:

·        Lines 189-190: In reduced tillage treatments, straw was incorporated manually on the soil surface. At what depth was the straw incorporated? Any further detail and description of this operation would be beneficial to interpret and understand data and results.

·        Lines 193-202: How the application of different amounts of fertilizers in the different cropping systems can affect the results? Any further detail and clarification on these aspects would be beneficial for the paper.

·        Line 244: A one m2 rice plants at maturity of each replication was harvested for yield determination. Is this area enough to properly determine yields in plot sizes of 15 m × 10 m? How was that one m2 identified withn the plots? Any further detail and clarification on these aspects would be beneficial for the paper.

·        Lines 249-258: can soil nutrient determination be influenced by fertilization practices (see lines 193-202)?

·        Figures 3 and 4 are not fully readable because of the used graphic design. We suggest adopting different scales or different design to make differences among different tillage treatments more significant.

·        Figures 4 and 5: we suggest inverting the numbering of figures since figure 5 appears before figure 4.

·        Lines 373-375: The highest cumulative N2O emission was produced by RoT + S in the rice-wheat cropping system whiles the 374 lowest was produced by RoT + S in the single rice cropping system. How can you explain these huge differences for the same tillage treatment? Any further detail and comment in the discussion would be beneficial for the paper.

Author Response

Reviewer 3

Comments and Suggestions for Authors

The methodology appears well developed and described, as well as the use of statistics is appropriate and accurate. References are rich and appropriate. The manuscript is clear and well structured. English language is correct and fluent. The paper is scientifically sound, and the experimental design appears generally appropriate. Figures/tables/images/schemes are appropriate and properly show the data. They are easy to be interpreted and understood, even if some of them could be re-edited in order to enhance legibility (see specific comments). Data is interpreted appropriately and consistently throughout the manuscript. Conclusions are consistent with the evidence and arguments presented.

Specific comments:

  1. Lines 189-190: In reduced tillage treatments, straw was incorporated manually on the soil surface. At what depth was the straw incorporated? Any further detail and description of this operation would be beneficial to interpret and understand data and results.

Reply: Thank you. Straw incorporation depth and further description has been inserted into the sentence. Lines 210 - Line 212 “In reduced tillage treatments, straw was incorporated manually on the soil surface at a depth of 3.0 cm – 5.0 cm, allowed to imbibe water and sink below the water surface.”

  1. Lines 193-202: How the application of different amounts of fertilizers in the different cropping systems can affect the results? Any further detail and clarification on these aspects would be beneficial for the paper.

Reply: Thank you. The application of different amounts of fertilizers in the different cropping systems was based on the locally recommended practice of high rice yield and frequency of cropping and the objective of our study was to compare the differences between different treatments in each location. For cropping system differences, maybe the fertilizer and soil properties will affect the results, but previous studies have state that the difference of methane mainly contributed by the temperature (Qian et al., 2022).

Qian, H., Zhang, N., Chen, J., Chen, C., Hungate, B. A., Ruan, J., Huang, S., Cheng, K., Song, Z., Hou, P., Zhang, B., Zhang, J., Wang, Z., Zhang, X., Li, G., Liu, Z., Wang, S., Zhou, G., Zhang, W., Ding, Y., Jiang, Y. (2022). Unexpected Parabolic Temperature Dependency of CH4 Emissions from Rice Paddies. Environmental Science & Technology, 56(8), 4871–4881. https://doi.org/10.1021/acs.est.2c00738

  1. Line 244: A one m2 rice plants at maturity of each replication was harvested for yield determination. Is this area enough to properly determine yields in plot sizes of 15 m × 10 m? How was that one m2 identified within the plots? Any further detail and clarification on these aspects would be beneficial for the paper.

Reply: Thank you for the comment. Further detail and clarification have been provided.

Line 269 – Line 272: “A three replicate of one m2 rice plants at maturity of each treatment replication was harvested using a one m2 quadrant. Rice plant that fell within the one m2 quadrant was harvested for yield determination”.

The one m2 area for sampling and yield determination was replicated three times per treatment and is enough for rice yield determination (Hindersah et al., 2022).

Hindersah, R., Kalay, A. Talahaturuson, A. (2022). Rice yield grown in different fertilizer combination and planting methods: Case study in Buru Island, Indonesia. Open Agriculture7(1), 871-881. https://doi.org/10.1515/opag-2022-0148

  1. Lines 249-258: can soil nutrient determination be influenced by fertilization practices (see lines 193-202)?

Reply: Thank you. The samples for soil properties determinations were taken after harvest. So the soil nutrient determination was not influenced by fertilization practice.

  1. Figures 3 and 4 are not fully readable because of the used graphic design. We suggest adopting different scales or different design to make differences among different tillage treatments more significant.

Reply: Thank you. The readability of the graph has been improved by adopting different scales and different design to make differences among different tillage treatments more significant.

  1. Figures 4 and 5: we suggest inverting the numbering of figures since figure 5 appears before figure 4.

Reply: Thank you. The correction has been duly effected in the manuscript.

  1. Lines 373-375: The highest cumulative N2O emission was produced by RoT + S in the rice-wheat cropping system whiles the 374 lowest was produced by RoT + S in the single rice cropping system. How can you explain these huge differences for the same tillage treatment? Any further detail and comment in the discussion would be beneficial for the paper. 

Reply: Thank you. The N2O emissions in paddy field during rice season mostly is lower significantly than dryland crop seasons (eg. Wheat, oil seed, etc), and the proportions of total emission contributed by N2O emissions in rice season is less than 15% (Linquist et al., 2015; Wang et al., 2016). Actually, under the flooded conditions during rice growth season, N2O emission was not stable, leading to slight differences in N2O concentration. So we deleted the descriptions.

Reference:

Linquist BA, Anders MM, Adviento-Borbe MA, Chaney RL, Nalley LL, da Rosa EF, van Kessel C (2015) Reducing greenhouse gas emissions, water use, and grain arsenic levels in rice systems. Glob Change Biol 21:407–417.

Wang W, Wu X, Chen A, Xie X, Wang Y, Yin C (2016) Mitigating effects of ex situ application of rice straw on CH4 and N2O emissions from paddy-upland coexisting system. Sci Rep 6:37402.